# Hydrodynamic and Thermodynamic Nonequilibrium Effects around Shock Waves: Based on a Discrete Boltzmann Method

**DOI:** 10.3390/e22121397

**Published:** 2020-12-10

**Authors:** Chuandong Lin, Xianli Su, Yudong Zhang

**Affiliations:** 1Sino-French Institute of Nuclear Engineering and Technology, Sun Yat-Sen University, Zhuhai 519082, China; linchd3@mail.sysu.edu.cn; 2Key Laboratory for Thermal Science and Power Engineering of Ministry of Education, Department of Energy and Power Engineering, Tsinghua University, Beijing 100084, China; 3School of Mechanics and Safety Engineering, Zhengzhou University, Zhengzhou 450001, China; ydzhang@zzu.edu.cn

**Keywords:** shock wave, velocity distribution function, nonequilibrium effect, discrete Boltzmann method

## Abstract

A shock wave that is characterized by sharp physical gradients always draws the medium out of equilibrium. In this work, both hydrodynamic and thermodynamic nonequilibrium effects around the shock wave are investigated using a discrete Boltzmann model. Via Chapman–Enskog analysis, the local equilibrium and nonequilibrium velocity distribution functions in one-, two-, and three-dimensional velocity space are recovered across the shock wave. Besides, the absolute and relative deviation degrees are defined in order to describe the departure of the fluid system from the equilibrium state. The local and global nonequilibrium effects, nonorganized energy, and nonorganized energy flux are also investigated. Moreover, the impacts of the relaxation frequency, Mach number, thermal conductivity, viscosity, and the specific heat ratio on the nonequilibrium behaviours around shock waves are studied. This work is helpful for a deeper understanding of the fine structures of shock wave and nonequilibrium statistical mechanics.

## 1. Introduction

Shock waves are widespread in many fields of physics and engineering [1,2,3,4], and their frequent occurrence is based on the fact that matter is more or less compressible [5]. The shock wave is a kind of disturbance waves that carries abundant energy and propagates at supersonic speed in the medium [6]. At the macroscopic level, it is a high-temperature, high-pressure, high-density surface moving forward at high-speed, through which the pressure, density, temperature, and other physical quantities of the medium undergo rapid changes [7,8]. Shock waves, as an objective phenomenon, are generated by lightning, earthquakes, volcanic eruptions, nuclear or chemical explosions, reentry vehicles, sonic boom of supersonic aircraft and any supersonic flying projectile, etc. [5,8].

In practice, the shock wave is extremely complex, as it covers a wide range of spatio-temporal scales, and incorporates hydrodynamic and thermodynamic nonequilibrium effects [9,10]. In fact, hydrodynamic and thermodynamic nonequilibrium effects often exert significant influences on fluid systems [11]. It is necessary to take these nonequilibrium effects into consideration in order to accurately predict fluid behaviours. To this aim, a possible method is to use molecular dynamics [12,13,14] or direct simulation Monte Carlo [15]. But both of them encounter the issue that the spatio-temporal scales amenable are rather limited due to the high computation cost. On the other side, traditional hydrodynamic methods which require less computation cost usually ignore more abundant, complex and essential thermodynamic nonequilibrium effects caused by interactions at microscopic scales [16,17]. The hydrodynamic models are based on the continuity assumption and usually lack the high physical accuracy for practice flows with sharp physical gradients and/or strong nonequilibrium effects. To address this problem, great efforts have been made and various kinetic models based on the Boltzmann equation have been developed.

Actually, there are two classes of kinetic models for fluid systems. One aims to provide a new algorithm to solving the hydrodynamic governing equations, e.g., Euler equations, incompressible or full Navier-Stokes equations. Such models include the lattice Boltzmann method [18,19,20,21] and the gas kinetic scheme [22,23]. These models could only simulate the fluid behaviours described by the original macroscopic fluid equations, they can not provide any information beyond the original equations. The other type of kinetic models is to capture both hydrodynamic and thermodynamic nonequilibrium behaviours beyond the macroscopic models. The successful physical models include the unified gas kinetic scheme [24,25,26], the discrete unified gas kinetic scheme [27,28], the discrete Boltzmann method (DBM) [29,30,31,32,33,34,35,36,37,38,39], etc. These powerful models are suitable for continuum and rarefied systems with a wide range of Knudsen numbers, and capable to subsonic and supersonic flows with essential nonequilibrium effects.

Compared with the unified gas kinetic scheme and the discrete unified gas kinetic scheme, the DBM requires less discrete velocities and distribution functions, hence it has a higher computational efficiency. From a physical modeling perspective, a DBM is approximately equivalent to a continuous fluid model plus a coarse-grained model of other relevant thermodynamic nonequilibrium effects. At present, the DBM has been developed as a nonequilibrium flow simulation tool for various flow systems, including the fluid instability [29,32,38], multiphase flow [34], shock and detonation [30,31,33,36,37,39]. Roughly speaking, the DBMs can be classed into two categories: the single-relaxation-time DBM [29,30,31,32,33,34,35,36] and the multiple-relaxation-time DBM [37,38,39]. In the single-relaxation-time model, there is only one relaxation time that controls a thermodynamic nonequilibrium system approaching its equilibrium state. In the multiple-relaxation-time DBM, the relaxation parameters for various kinetic modes are independent of each other. Actually, the multiple-relaxation-time DBM reduces to a single-relaxation-time model if all relaxation parameters equal to each other.

In this article, we use the multiple-relaxation-time DBM to study nonequilibrium shock waves, where both specific heat ratio and Prandtl number are adjustable [39]. Thanks to its kinetic nature, the versatile DBM can be adopted to capture and measure both hydrodynamic and thermodynamic nonequilibrium behaviours in an accurate and effective way [39]. The DBM is briefly introduced in Appendix A. The rest of the paper is structured as follows. The study of local nonequilibrium effects is presented in Section 2. Then the investigation of global nonequilibrium effects is shown in Section 3. Finally, Section 4 concludes.

## 2. Local Nonequilibrium Effects

It is noteworthy that the DBM has the capability of measuring the nonequilibrium manifestations in subsonic, sonic and supersonic flows accurately [29,30,31,32,33,34,35,36,37,38,39]. Actually, the DBM is capable to recover the velocity distribution function in two ways. One is to obtain the main features of the velocity distribution function by analysis of the detailed nonequilibrium quantity ${\hat{f}}_{i}^{neq}$ that has specific physical meanings. For example, ${\hat{f}}_{5}^{neq}={\hat{f}}_{5}-{\hat{f}}_{5}^{eq}$ denotes twice the nonorganized energy in the *x* direction; ${\hat{f}}_{7}^{neq}={\hat{f}}_{7}-{\hat{f}}_{7}^{eq}$ represents twice the nonorganized energy in the *y* direction; ${\hat{f}}_{8}^{neq}={\hat{f}}_{8}-{\hat{f}}_{8}^{eq}$ stands for twice the nonorganized energy flux in the *x* direction; ${\hat{f}}_{9}^{neq}={\hat{f}}_{9}-{\hat{f}}_{9}^{eq}$ is twice the nonorganized energy flux in the *y* direction. In previous works [29,30,40], the DBM has already been utilized to get the characteristics of the velocity distribution function around the detonation wave, shock front, rarefaction wave, and material interface. The other is to recover the main characteristics of the velocity distribution quantitatively through the macroscopic quantities and their spatial and temporal derivatives, which can be achieved by the Chapman-Enskog expansion [35].

### 2.1. Nonequilibrium Manifestations

To have a better understanding of the nonequilibrium effects upon the fine structure of the shock wave, we probe the nonequilibrium variables, effects, and degrees in this subsection. Note that the main characteristics of the velocity distribution function can be obtained from the thermodynamic nonequilibrium manifestations [29,30,40].

Now we consider a shock wave propagating from left to right. At the beginning, the shock front is located at x=0.01 with the Mach number Ma=2. The computation is carried out with the mesh number $N_{x}\times N_{y}$ = 10,000 × 1. The spatial step is $\Delta x=\Delta y=10^{-4}$, the temporal step $\Delta t=10^{-5}$, and relaxation frequency $S_{i}=10^{3}$. The initial state is set by the Hugoniot relation,
$$\left\{\begin{array}{c} {\left(\rho ,u_{x},u_{y},T,P\right)}_{L}=\left(2.667,1.479,0,1.688,4.5\right),\\ {\left(\rho ,u_{x},u_{y},T,P\right)}_{R}=\left(1,0,0,1,1\right), \end{array}\right.$$
where the subscripts *L* and *R* denote 0<x≤0.01 and 0.01<x≤1, respectively. In addition, inflow and outflow boundary conditions are employed in the *x* direction, and the periodic boundary condition is adopted in the *y* direction.

The profiles of density ρ, pressure *P*, temperature *T*, and horizontal velocity $u_{x}$ near the shock front are illustrated in Figure 1a. The solid lines with squares, circles, upper triangles and lower triangles represent the density, pressure, temperature, and horizontal velocity, respectively. Three vertical dashed lines are plotted to guide the locations at $x_{1}=0.9545,x_{2}=0.9575$ and $x_{3}=0.9605$, respectively. It’s clear that these quantities increase dramatically as the shock wave travels forward through the medium.

Figure 1b displays the gradients of the physical quantities. The solid lines with squares, circles, upper triangles and lower triangles denote the gradients of density, pressure, temperature and horizontal velocity, respectively. Obviously, these physical gradients are quite distinguishable from each other with different amplitudes and trough locations around the wavefront. The minima of ρ, *P*, *T* and $u_{x}$ are located at x=0.95705, 0.95755, 0.95955 and 0.95885, respectively.

Figure 1c describes the nonorganized energy ${\hat{f}}_{5}^{neq}$ and nonorganized energy flux in the x direction ${\hat{f}}_{8}^{neq}$. The solid lines with circles and squares indicate ${\hat{f}}_{5}^{neq}$ and ${\hat{f}}_{8}^{neq}$, respectively. It is evident that both of them are greater than zero, and they first increase and then decrease around the wavefront. Their peaks nearly coincide at $x_{2}=0.9575$. Physically, ${\hat{f}}_{5}^{neq}$ and ${\hat{f}}_{8}^{neq}$ are related to the gradients of fluid velocity and temperature (∇u and ∇T), respectively [41]. As shown in Figure 1b, the troughs of the physical gradients $\nabla u_{x}$ and ∇T are close to each other. This is the reason why the locations of their minima are near.

Figure 1d shows the profiles of nonequilibrium effect (Φ) and deviation degrees ($\Delta_{r}$ and $\Delta_{a}$). The line with squares denotes the nonequilibrium effect,
(1)$$\Phi =\sqrt{\sum {\left({\hat{f}}_{i}^{neq}\right)}^{2}}.$$
The solid lines with upper triangles and circles display the absolute and relative deviation degrees defined as,
(2)$$\Delta_{a}=\int_{-\infty }^{+\infty }\int_{-\infty }^{+\infty }\int_{-\infty }^{+\infty }\left|f-f^{eq}\right|dv_{x}dv_{y}d\eta ,$$
and
(3)$$\Delta_{r}=\frac{\int_{-\infty }^{+\infty }\int_{-\infty }^{+\infty }\int_{-\infty }^{+\infty }\left|f-f^{eq}\right|dv_{x}dv_{y}d\eta }{\int_{-\infty }^{+\infty }\int_{-\infty }^{+\infty }\int_{-\infty }^{+\infty }\left|f+f^{eq}\right|dv_{x}dv_{y}d\eta }=\frac{1}{2\rho }\int_{-\infty }^{+\infty }\int_{-\infty }^{+\infty }\int_{-\infty }^{+\infty }\left|f-f^{eq}\right|dv_{x}dv_{y}d\eta =\frac{\Delta_{a}}{2\rho },$$
respectively. Obviously, $\Delta_{a}$, $\Delta_{r}$ and Φ first grow, then decrease and form peaks near the wavefront. The peak of Φ nearly coincides with those of ${\hat{f}}_{5}^{neq}$ and ${\hat{f}}_{8}^{neq}$ at $x_{2}=0.9575$, because all the three nonequilibrium quantities Φ, ${\hat{f}}_{5}^{neq}$ and ${\hat{f}}_{8}^{neq}$ depend upon the gradients of fluid velocity and temperature. The peak of $\Delta_{r}$ is closer to the peak of Φ, and the peak of $\Delta_{a}$ is located behind them. The hysteresis of this peak arises from the change in density as it passes through the shock wave.

### 2.2. Recovery of Velocity Distribution Function

Next, we derive the formula of the velocity distribution function through the macroscopic quantities and their spatial and temporal derivatives for the multiple-relaxation-time DBM. Compared with the derivation in Ref. [35], the specific heat ratio is under consideration here. Then the equilibrium and nonequilibrium velocity distribution functions are recovered and analyzed at three different locations around the shock front that propagates forward.

Let us consider the case with the Prandtl number Pr=1, which means the relaxation frequency $S_{i}$ equals to each other in the multiple-relaxation-time discrete Boltzmann model. Additionally, the relaxation time is defined as $\tau_{i}=1/S_{i}$. Via the Chapman–Enskog analysis, the first order approximation of the velocity distribution function *f* can be obtained, as below
(4)$$\begin{array}{c} f\approx f^{eq}+f^{\left(1\right)}\\ =f^{eq}\left\{1-\tau_{i}\left[D_{\rho }\left(\frac{\partial \rho }{\partial t}+v_{\alpha }\frac{\partial \rho }{\partial r_{\alpha }}\right)+D_{T}\left(\frac{\partial T}{\partial t}+v_{\alpha }\frac{\partial T}{\partial r_{\alpha }}\right)+D_{u_{\beta }}\left(\frac{\partial u_{\beta }}{\partial t}+v_{\alpha }\frac{\partial u_{\beta }}{\partial r_{\alpha }}\right)\right]\right\}, \end{array}$$
with the equilibrium distribution function
(5)$$f^{eq}=\frac{\rho }{2\pi T}{\left(\frac{1}{2\pi IT}\right)}^{1/2}\exp\left[-\frac{\left(v-u\right)\cdot \left(v-u\right)}{2T}-\frac{\eta^{2}}{2IT}\right],$$
where $D_{\rho }=1/\rho$, $D_{T}=\left[-\left(D+1\right)/2T+{\left|v_{\alpha }-u_{\alpha }\right|}^{2}/2T^{2}+\eta^{2}/2IT^{2}\right]$, $D_{u_{\beta }}=\left(v_{\beta }-u_{\beta }\right)/T$, D=2 stands for the space dimension, *I* indicates extra degrees of freedom due to vibration and/or rotation, and η corresponds to vibrational and/or rotational energies. Appendix B provides the detailed derivation process.

Figure 2 presents the velocity distribution functions. To quantitatively study the characteristics of the distribution functions, three-dimensional (3D) to one-dimensional (1D) velocity distribution functions at each position are drawn. The three columns depict velocity distribution functions at locations $x_{1}=0.9545,x_{2}=0.9575,x_{3}=0.9605$ from left to right, respectively. Figure 2(a1–c1) intuitively show the 3D distribution functions in velocity space $\left(v_{x},v_{y}\right)$ at different points. Figure 2(a2–c2) plot the two-dimensional (2D) contours of distribution functions. Figure 2(a3–c3) are the 1D distribution functions, where the solid and dashed lines represent the velocity distribution function $f\left(v_{x}\right)=\;\int \int \;fdv_{y}d\eta$ and its equilibrium counterpart $f^{eq}\left(v_{x}\right)=\;\int \int \;f^{eq}dv_{y}d\eta$, respectively. Figure 2(a4–c4) display the other 1D profiles of the distribution functions, where the solid lines denote $f\left(v_{y}\right)=\;\int \int \;fdv_{x}d\eta$ and the dashed lines indicate $f^{eq}\left(v_{y}\right)=\;\int \int \;f^{eq}dv_{x}d\eta$. From Figure 2, we can obtain the following points.

(I) There are clear peaks in the Figure 2(a1–c1). From left to right, the peaks get “smaller” and more “pointy”. Because the shock wave passes through the medium, the physical quantities increase rapidly. The velocity distribution functions become progressively sharper from post-wave to pre-wave.

(II) The contours presented in Figure 2(a2–c2) are symmetrical in the $v_{y}$ direction and asymmetric in the $v_{x}$ direction at various locations. From left to right, the contours gradually look like eggs due to the nonequilibrium effects. The reason is that the system is periodic in the *y* direction and there is flow or flux just in the *x* direction.

(III) As Figure 2(a3–c3) show, $f^{eq}\left(v_{x}\right)$ is symmetric and the symmetry axis is $u_{x}$, while $f(v_{x})$ is asymmetric due to the nonequilibrium effects. The velocity distribution function $f(v_{x})$ is “lower” and “fatter” than the equilibrium velocity distribution function $f^{eq}\left(v_{x}\right)$, which indicates the nonorganized energy $f_{5}^{neq}>0$ (see Figure 1). As $v_{x}$ deviates far from $u_{x}$, the distribution function $f(v_{x})$ for $v_{x}>u_{x}$ is larger than the corresponding value of $f(v_{x})$ for $v_{x}<u_{x}$, which indicates ${\hat{f}}_{8}^{neq}>0$ (see Figure 1).

(IV) Both $f(v_{y})$ and $f^{eq}\left(v_{y}\right)$ in Figure 2(a4–c4) are symmetrical and the axes of symmetry are $u_{y}=0$, because there is no flow or flux in the *y* direction. Additionally, the velocity distribution function $f(v_{y})$ “higher” and “thinner” than the equilibrium velocity distribution function $f^{eq}\left(v_{y}\right)$. It indicates ${\hat{f}}_{7}^{neq}<0$ and ${\hat{f}}_{9}^{neq}=0$ (which is not shown here).

(V) In the last two rows, the areas of $f(v_{x})$, $f(v_{y})$, $f^{eq}\left(v_{x}\right)$, and $f^{eq}\left(v_{y}\right)$ decrease gradually from left to right. In fact, the area corresponds to density ρ, which increases as the shock wave travels from left to right. The widths of $f(v_{x})$, $f(v_{y})$, $f^{eq}\left(v_{x}\right)$, and $f^{eq}\left(v_{y}\right)$ are related to the temperature *T*, which increases from the first to the third columns. As a result, the curves become “higher” and “leaner”.

It should be mentioned that the numerical results presented in Figure 1 are consistent with the analytic solutions in Figure 2. Up to this point, from Figure 1 and Figure 2, we can obtain an intuitive understanding of the deviation between the nonequilibrium and equilibrium velocity distribution functions around the shock wave. More simulations are conducted in the following section in order to have a better understanding of the physical mechanisms of the nonequilibrium shock waves.

## 3. Global Nonequilibrium Effects

We define the following nonequilibrium quantities to further study the nonequilibrium effects around the shock wave,
(6)$$\Psi =\;\int \int \;\sqrt{\sum {\left({\hat{f}}_{i}^{neq}\right)}^{2}}dxdy,$$
(7)$$\Psi_{2,xx}=\;\int \int \;{\hat{f}}_{5}^{neq}dxdy,$$
(8)$$\Psi_{3,1,x}=\;\int \int \;{\hat{f}}_{8}^{neq}dxdy,$$
which denote the global nonequilibrium effects, the global nonorganized energy in the *x* direction, and the global nonorganized energy flux in the *x* direction, respectively. Next, the DBM is employed to study the effects of the relaxation frequency, Mach number, thermal conductivity, viscosity, and specific heat ratio on those global nonequilibrium variables around shock waves, respectively.

### 3.1. Impact of the Relaxation Frequency

In the physical sciences, relaxation usually means the return of a perturbed system into equilibrium. In our simulations, the relaxation frequency $S_{i}$ governs the speed of relaxation process from a nonequilibrium state to its equilibrium state. It plays a key role in the nonequilibrium manifestations. Here, let us investigate the nonequilibrium effects around the shock wave under different relaxation frequencies. The relaxation frequencies are chosen as $S_{i}=1\times 10^{3}$, $2\times 10^{3}$, $4\times 10^{3}$, $8\times 10^{3}$, $1.6\times 10^{4}$, and $3.2\times 10^{4}$, respectively.

First of all, to give comparisons between Riemann solutions and DBM results, the profiles of density ρ, horizontal velocity $u_{x}$, pressure *P*, and temperature *T* near the shock front are shown in Figure 3. The black solid line represents the result of the Riemann solutions, and the other lines stand for the DBM results under various relaxation frequencies. It is evident that the Riemann solutions show a straight vertical line near the shock wave, while the DBM results display a smooth curve around the shock front. Because the DBM contains viscosity and thermal conductivity as well as other detailed thermodynamic nonequilibrium effects, while the Riemann solutions are based on the Euler equations that ignore all thermodynamic nonequilibrium effects. As the relaxation frequency increases, the nonequilibrium effects decrease, and the DBM results are close to the Riemann solutions.

Figure 4a displays the profiles of the nonequilibrium effect Φ with various relaxation frequencies at time t=0.4. With the increasing relaxation frequency, the amplitude of the nonequilibrium effect remains constant, while the width of the nonequilibrium region increases proportionally. Figure 4b–d illustrate the global nonequilibrium quantities lnΨ, $\ln\Psi_{2,xx}$, and $\ln\Psi_{3,1,x}$ versus $\ln S_{i}$. Circles, triangles, and squares denote the simulation results, and the lines represent the fitting equations: $\ln\Psi =2.892-2\times \ln S_{i}$, $\ln\Psi_{2,xx}=-0.423-2\times \ln S_{i}$ and $\ln\Psi_{3,1,x}=1.008-2\times \ln S_{i}$, respectively. It is evident that the fitting results agree well with the simulation solutions, and $\ln\Psi ,\ln\Psi_{2,xx}$ and $\ln\Psi_{3,1,x}$ decrease as $\ln S_{i}$ increases linearly. Physically, with the increasing relaxation frequency, the relaxation time from $\hat{f_{i}}$ to ${\hat{f}}_{i}^{eq}$ decreases, and the nonequilibrium effects around the shock wave reduce [31].

### 3.2. Impact of the Mach Number

Mach number Ma is a dimensionless quantity in fluid dynamics that represents the ratio of flow velocity past a boundary to the local speed of sound [42]. Now, let us demonstrate the relationship between the nonequilibrium effects and Mach number. We simulate a shock wave with various Mach numbers from Ma=1.5 to 5 at time t=0.4, with Pr=1. The Hugoniot relation sets the initial states in all cases.

Figure 5 exhibits the nonequilibrium effects versus the Mach numbers. Figure 5a–c are for the global nonequilibrium effects, the global nonorganized energy in the *x* direction, and the global nonorganized energy flux in the *x* direction, respectively. The symbols denote the simulation results and the lines are for the fitting results. The fitting equations are lnΨ=−14.557+1.724×Ma, $\Psi_{2,xx}=\left[0.635\times \exp\left(\mathrm{Ma}/1.65\right)-1.45\right]\times 10^{-6}$, and $\Psi_{3,1,x}=\left[1.778\times \exp\left(\mathrm{Ma}/1.13\right)-7.721\right]\times 10^{-6}$, respectively. Clearly, as the Mach number increases, the global nonequilibrium effects lnΨ increases in a linear relationship approximately, while the global nonorganized energy $\Psi_{2,xx}$ and energy flux in the *x* direction $\Psi_{3,1,x}$ grow exponentially. With the increasing Mach number, both physical quantities and physical gradients increase around the shock wave, which leads to increasing nonequilibrium effects [31].

### 3.3. Impact of the Thermal Conductivity

Next, the simulation of the shock wave is carried out for various values of the thermal conductivity. In the DBM, the thermal conductivity takes the form $\kappa =\left(D+I+2\right)P/\left(2S_{\kappa }\right)$ in terms of the pressure P=ρT and the parameter $S_{\kappa }=S_{8}=S_{9}$. Clearly, the thermal conductivity changes due to the variable pressure across the shock wave. For convenience of quantitative study, we choose the pressure $P=(P_{L}+P_{R})/2$ where $P_{L}=4.5$ and $P_{R}=1.0$ denote the pressure on the left and right sides of the shock wave, respectively. Besides, the collision parameters are $S_{i}=2\times 10^{3}$, except $S_{8}=S_{9}=0.5\times 10^{3}$, $1\times 10^{3}$, $2\times 10^{3}$, $3\times 10^{3}$, $4\times 10^{3}$, $5\times 10^{3}$, $6\times 10^{3}$, $7\times 10^{3}$, and $8\times 10^{3}$, for the nine different cases. Correspondingly, the thermal conductivity is κ=0.0165, 0.00825, 0.004125, 0.00275, 0.0020625, 0.00165, 0.001375, 0.001178571, and 0.00103125, respectively.

Figure 6 displays the simulation results and the fitting solutions. The fitting equations are $\Psi =\left[3.144\times \exp\left(-0.0182/\kappa \right)+3.081\right]\times 10^{-6}$ and $\Psi_{3,1,x}=\left(0.246+103.426\kappa \right)\times 10^{-6}$, respectively. Obviously, with the increasing value of 1/κ, the nonequilibrium effect Ψ decreases exponentially and it tends to be a constant, and $\Psi_{3,1,x}$ grows linearly with the thermal conductivity. Physically, with the increasing $S_{\kappa }$, the thermal conductivity reduces, the nonorganized energy flux decreases, and the nonequilibrium effects become week [41].

### 3.4. Impact of the Viscosity

In this subsection, we simulate a 1D steady shock wave with various values of viscosity. In the DBM, the dynamic viscosity is expressed by $\mu =P/S_{\mu }$ with the parameter $S_{\mu }=S_{5}=S_{6}=S_{7}$. Similarly, the dynamic viscosity is variable around the shock front. For convenience, we adopt $P=(P_{L}+P_{R})/2$ at the shock wave. The collision parameters are $S_{\mu }=2.5\times 10^{3}$, $3\times 10^{3}$, $4\times 10^{3}$, $5\times 10^{3}$, $6\times 10^{3}$, $7\times 10^{3}$, $8\times 10^{3}$, $9\times 10^{3}$ and $1\times 10^{4}$, and $S_{i}=5\times 10^{3}$ for the others. The corresponding dynamic viscosity is μ = 0.0011, 0.000916667, 0.0006875, 0.00055, 0.000458333, 0.000392857, 0.00034375, 0.000305556, and 0.000275, respectively.

Figure 7 displays the simulation and fitting results: (a) the global nonequilibrium effects versus the reciprocal of μ and (b) the global nonorganized energy in the *x* direction versus μ. The circles and triangles stand for the DBM results and the solid lines stand for the solutions of the fitting equations: $\Psi =\left[1.378\times \exp(-0.0025/\mu )+1.177\right]\times 10^{-6}$, $\Psi_{2,xx}=\left(-2.7\times 10^{-6}+1.1937\mu \right)\times 10^{-4}$.

It can be found that the global nonequilibrium effect Ψ decreases and approaches a constant with the increasing 1/μ, and there is a linear growth relationship between $\Psi_{2,xx}$ and μ. Physically, with the increasing $S_{\mu }$, the viscosity decreases, the viscous shear reduces, the nonorganized energy decreases, and the nonequilibrium effects become smaller [41].

### 3.5. Impact of the Specific Heat Ratio

In thermal physics and thermodynamics, the specific heat ratio is the ratio of the heat capacity at constant pressure to heat capacity at constant volume. In this subsection, the simulation of shock wave is carried out for different specific heat ratios. The specific heat ratio is γ=(D+I+2)/(D+I). In our simulations, the specific heat ratio γ is adjusted by varying the extra degrees of freedom *I*. The extra degrees of freedom are I=7,6,5,4,3,2,1, and 0, and the corresponding specific heat ratios are γ=1.22,1.25,1.29,1.33,1.4,1.5,1.67, and 2, respectively.

Figure 8 shows the simulation and fitting results. Triangles and squares represent the simulation results, and the lines denote the fitting equations: $\Psi_{2,xx}=\left(2.876-0.899\gamma \right)\times 10^{-7}$ and $\Psi_{3,1,x}=\left[-14.215\times \exp\left(-\gamma /0.395\right)+7.127\right]\times 10^{-7}$. Clearly, as the specific heat ratio increases, the global nonorganized energy $\Psi_{2,xx}$ decreases in a linear ship approximately, while the global nonorganized energy flux $\Psi_{3,1,x}$ increases in an exponential form. The nonorganized energy in the *x* direction ${\hat{f}}_{5}^{neq}$ is the function of the fluid velocity and extra degrees of freedom [41]. The fluid velocity is similar in the simulations with different specific heat ratios. Meanwhile, as the extra degrees of freedom increases, the specific heat ratio decreases, and ${\hat{f}}_{5}^{neq}$ increases. So the decreasing extra degrees of freedom result in a decrease of $\Psi_{2,xx}$. Additionally, ${\hat{f}}_{8}^{neq}$ is related to the gradient of temperature and thermal conductivity. Consequently, with the increases of extra degrees of freedom, the specific heat ratio decreases, the temperature and its gradient decrease, and the thermal conductivity increases. The decreasing temperature gradient and increasing thermal conductivity have opposite impacts on the nonorganized energy flux, and temperature gradient plays a dominant role. So the increasing specific heat ratio result in an increase of $\Psi_{3,1,x}$ [41].

## 4. Conclusions

Shock wave is relevant to many fields of physics and engineering. When the shock wave passes through the medium, it draws the system out of equilibrium. In this article, we use the multiple-relaxation-time DBM in order to study hydrodynamic and thermodynamic nonequilibrium effects around the shock wave. Via the Chapman-Enskog analysis, we derive the formula of the velocity distribution function through the macroscopic quantities and their spatial and temporal derivatives with the flexible specific heat ratio. Moreover, we probe the nonequilibrium variables and effects in order to have a better understanding of the nonequilibrium effects upon the fine structure of the shock wave. In addition, to give an intuitive description, we define the absolute deviation degree $\Delta_{a}$ and relative deviation degree $\Delta_{r}$ in order to describe the departure of velocity distribution function from its equilibrium counterpart. It is found that the deviation degrees and nonequilibrium effects first grow, then decrease, and form peaks at the shock wave. Additionally, there are small or remarkable distances among those peaks, because the nonequilibrium behaviours are associated with the physical gradients whose peaks are near or far from each other around the shock wave.

We illustrate 1D, 2D, and 3D velocity distribution functions at three different locations around the shock front that propagates forward in the *x* direction in order to quantitatively study the characteristics of the distribution functions. The 3D velocity distribution functions become progressively sharper from post-wave to pre-wave. The 2D velocity distribution functions near the shock wave are symmetrical in the $v_{y}$ direction and asymmetrical in the $v_{x}$ direction. The contours look like eggs gradually from left to right due to the nonequilibrium effects. As for the 1D velocity distribution functions, the areas of $f(v_{x})$, $f(v_{y})$, $f^{eq}\left(v_{x}\right)$, and $f^{eq}\left(v_{y}\right)$ correspond to density ρ, and they gradually increase as the shock wave travels from left to right. Meanwhile, the widths of $f(v_{x})$, $f(v_{y})$, $f^{eq}\left(v_{x}\right)$, and $f^{eq}\left(v_{y}\right)$ that are related to the temperature *T* increases step by step when the shock wave propagates forward. As a result, the 1D velocity distribution function curves become “higher” and “leaner” from post-wave to pre-wave.

Besides, comparisons are made between the Riemann solutions and DBM results. Physically, the DBM contains viscosity and thermal conductivity, as well as other detailed thermodynamic nonequilibrium effects, while the Riemann solutions are based on the Euler equations, which ignore all of the thermodynamic nonequilibrium effects. As the relaxation frequency increases, the nonequilibrium effects decrease, and the DBM results become close to the Riemann solutions. Moreover, we study the impacts of relaxation frequency, Mach number, thermal conductivity, viscosity, and specific heat ratio upon the nonequilibrium effects. The nonequilibrium quantities under consideration include the global nonequilibrium effects Ψ, the global nonorganized energy in the *x* direction $\Psi_{2,xx}$ and the global nonorganized energy flux in the *x* direction $\Psi_{3,1,x}$. From the numerical simulations and corresponding fitting equations, the following points can be obtained: (I) With the increasing relaxation frequency, the amplitude of the nonequilibrium effect remains constant, while the width of the nonequilibrium region increases proportionally. Additionally, $\ln\Psi ,\ln\Psi_{2,xx}$ and $\ln\Psi_{3,1,x}$ decrease as $\ln S_{i}$ increases linearly. (II) As the Mach number increases, all of the nonequilibrium effects increase, the global nonequilibrium effects Ψ increases in a linear relationship approximately, while the global nonorganized energy $\Psi_{2,xx}$ and global nonorganized energy flux in the *x* direction $\Psi_{3,1,x}$ grow exponentially. (III) With the decrease of the thermal conductivity, the global nonequilibrium effects Ψ gradually decreases and tends to be a constant, and $\Psi_{3,1,x}$ grows linearly with the increasing κ. (IV) With the decrease of viscosity, the global nonequilibrium effects Ψ decreases gradually and then approaches a constant, and there is a linear growth relationship between $\Psi_{2,xx}$ and μ. (V) As the specific heat ratio increases, the global nonorganized energy $\Psi_{2,xx}$ decreases in a linear ship, while the nonorganized energy flux $\Psi_{3,1,x}$ increases in an exponential form. The physical mechanisms are analyzed and explained for the above phenomena. This work is helpful for obtaining a deeper understanding of the fine structures of shock waves and the nonequilibrium statistical mechanics.

## Figures and Tables

**Figure 1 entropy-22-01397-f001:**
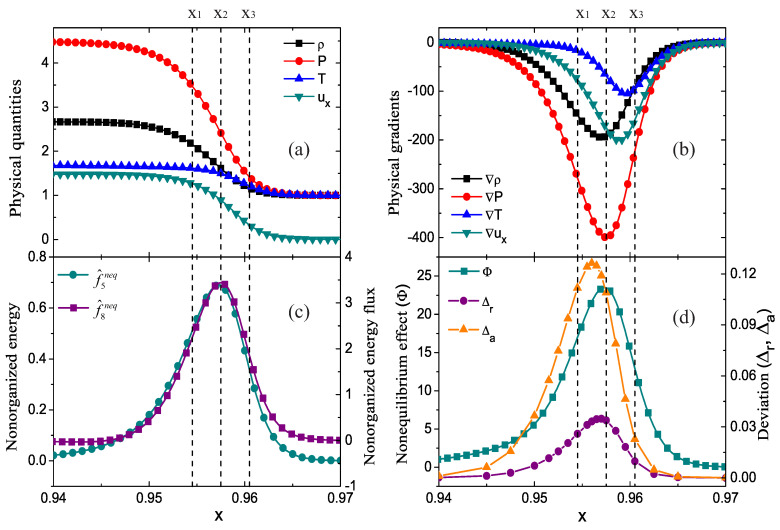
Profiles of physical quantities (ρ, *P*, *T*, $u_{x}$) (**a**), physical gradients (**b**), nonorganized energy ${\hat{f}}_{5}^{neq}$ and energy flux in the x direction ${\hat{f}}_{8}^{neq}$ (**c**), nonequilibrium effect Φ and deviation degrees (relative deviation degree $\Delta_{r}$, absolute deviation degree $\Delta_{a}$) (**d**) near the wavefront.

**Figure 2 entropy-22-01397-f002:**
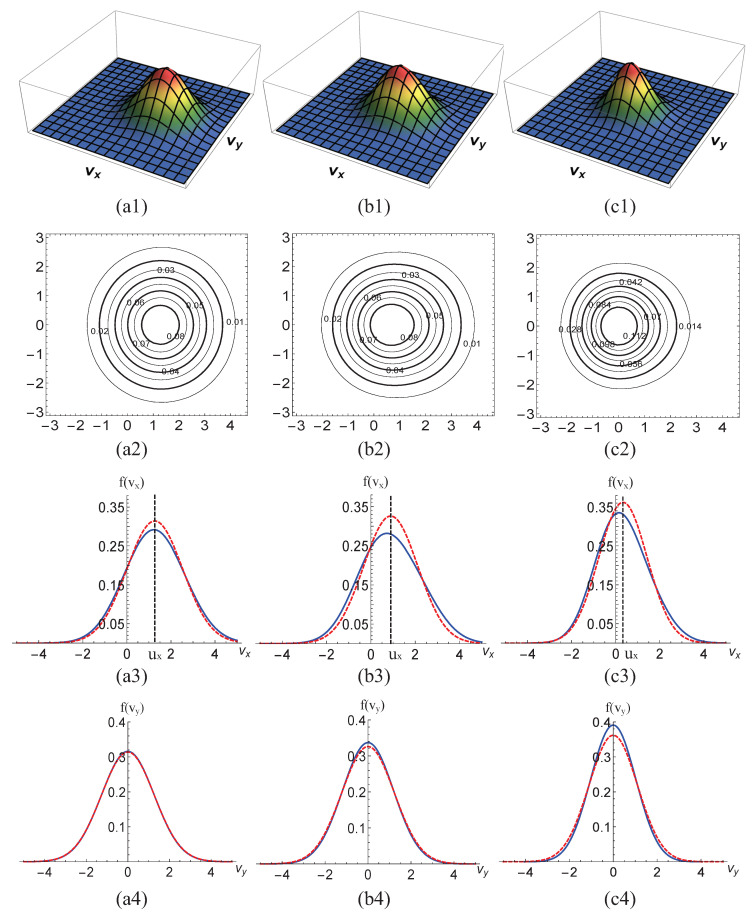
The first row shows the 3D velocity distribution functions, the second row plots the 2D velocity distribution functions, and in the last two rows the solid lines represent $f(v_{x})$ or $f(v_{y})$ and the dashed lines denote $f^{eq}\left(v_{x}\right)$ or $f^{eq}\left(v_{y}\right)$. The three columns from left to right depict the velocity distribution functions at locations $x_{1}=0.9545,x_{2}=0.9575$ and $x_{3}=0.9605$, respectively.

**Figure 3 entropy-22-01397-f003:**
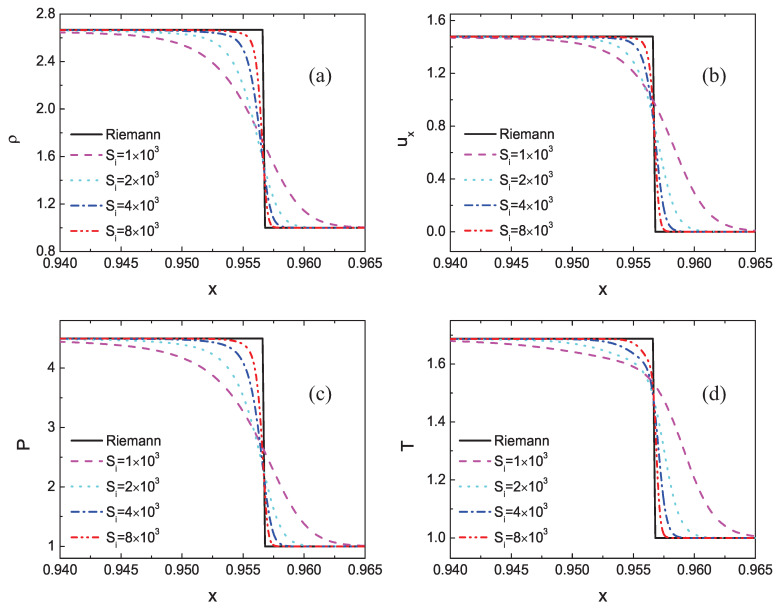
Comparisons between Riemann solutions and DBM results under various relaxation frequencies: (**a**) density, (**b**) horizontal velocity, (**c**) pressure, and (**d**) temperature.

**Figure 4 entropy-22-01397-f004:**
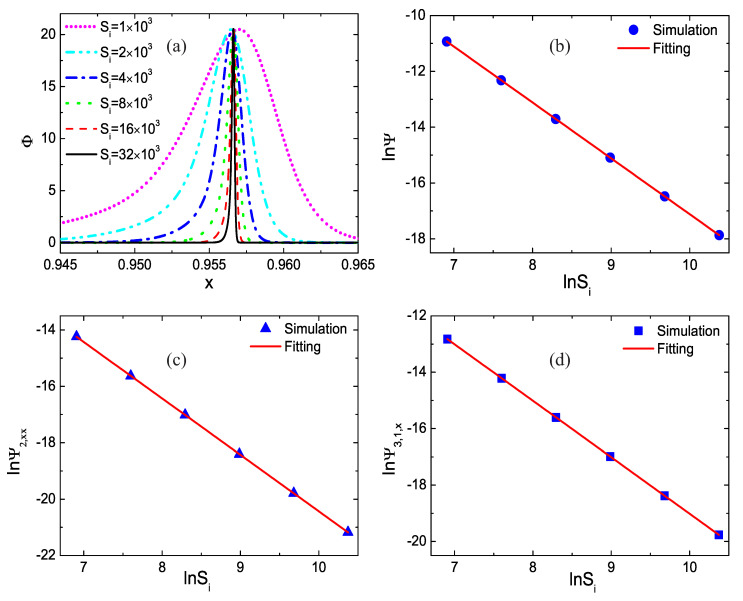
Nonequilibrium effects with various relaxation frequencies: (**a**) nonequilibrium effect Φ, (**b**) the global nonequilibrium effects Ψ, (**c**) the global nonorganized energy in the *x* direction $\Psi_{2,xx}$, and (**d**) the global nonorganized energy flux in the *x* direction $\Psi_{3,1,x}$.

**Figure 5 entropy-22-01397-f005:**
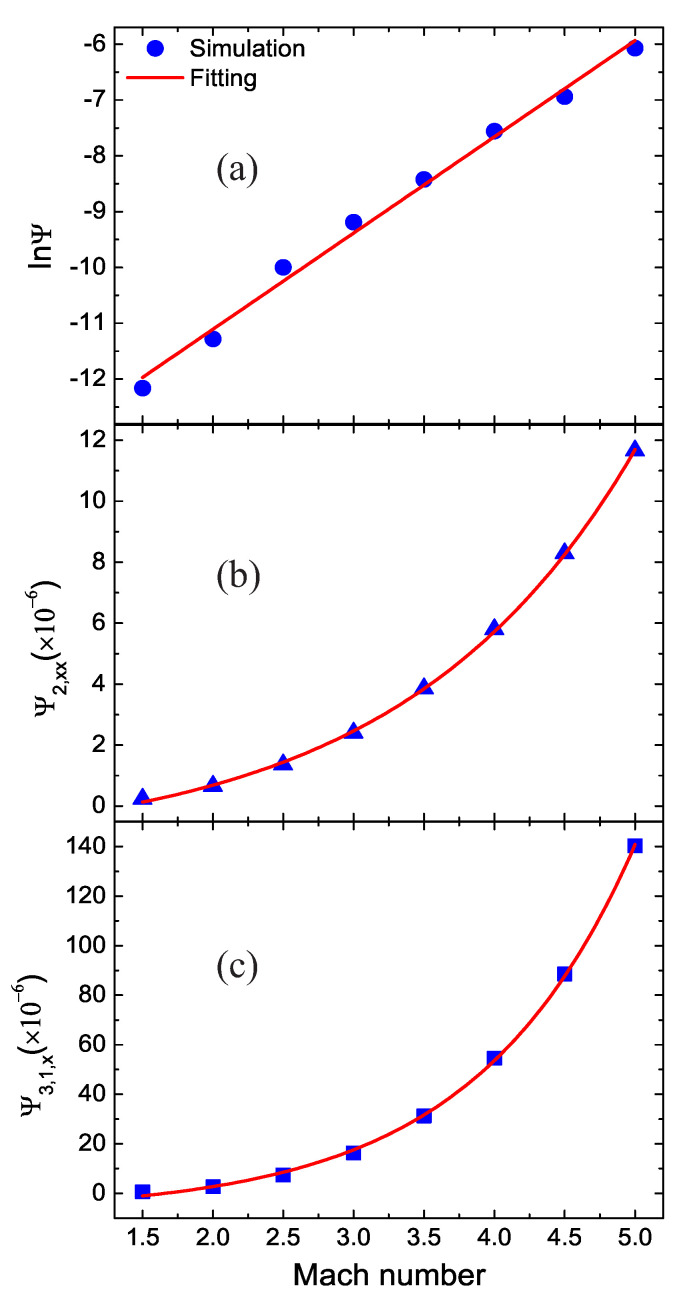
Nonequilibrium effects with various Mach numbers: (**a**) the global nonequilibrium effects Ψ, (**b**) the global nonorganized energy in the *x* direction $\Psi_{2,xx}$, and (**c**) the global nonorganized energy flux in the *x* direction $\Psi_{3,1,x}$.

**Figure 6 entropy-22-01397-f006:**
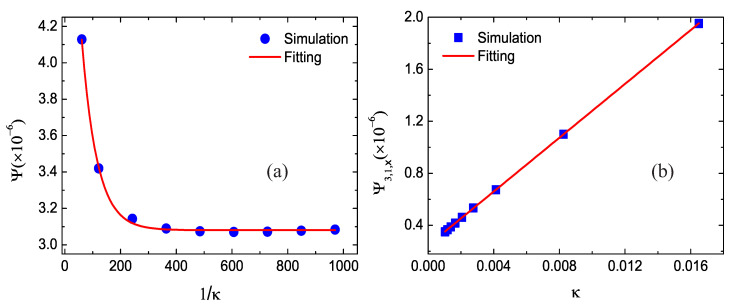
Nonequilibrium effects with different thermal conductivity: (**a**) the global nonequilibrium effects Ψ versus 1/κ and (**b**) the global nonorganized energy flux $\Psi_{3,1,x}$ in the *x* direction versus κ.

**Figure 7 entropy-22-01397-f007:**
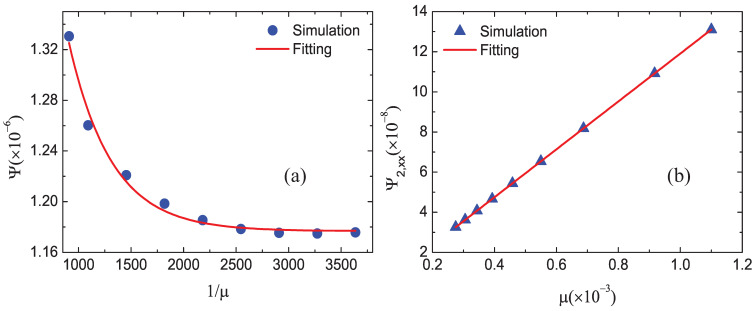
Nonequilibrium effects with various viscosity: (**a**) the global nonequilibrium effects Ψ versus 1/μ and (**b**) the global nonorganized energy $\Psi_{2,xx}$ in the *x* direction versus μ.

**Figure 8 entropy-22-01397-f008:**
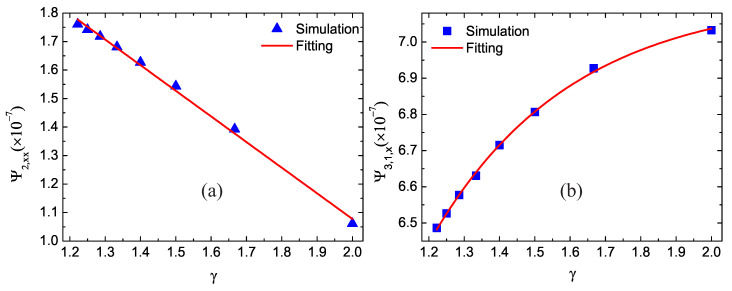
Nonequilibrium effects with various specific heat ratios: (**a**) the global nonorganized energy $\Psi_{2,xx}$ in the *x* direction and (**b**) the global nonorganized energy flux $\Psi_{3,1,x}$ in the *x* direction.

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
