# Peer review of "Hydrodynamic and Thermodynamic Nonequilibrium Effects around Shock Waves: Based on a Discrete Boltzmann Method"

_entropy, 2020, doi:10.3390/e22121397_

Round 1
Reviewer 1 Report
The paper deals with the study of shock waves in terms of the so called discrete Boltzmann method. The results reported in the manuscript suggest that the authors have produced something that could be of the interest of the readers of Entropy. However, the structure of the manuscript exhibits some faults in the exposition of ideas that make no possible to provide some words about the novelty and soundness of the results. Therefore, a major review is necessary before considering the paper for publication.
Some guidelines are suggested below, but the items are not exhaustive.
1.- I strongly recommend avoiding the excessive use of acronyms throughout the body of the manuscript. I understand that it is implemented to circumvent repetition, however, too many acronyms used in the same paragraph achieve the opposite effect because they obscure the presentation of ideas. For instance, the acronym HTNEs is used three times in lines-35-37, paragraph 2, page 2. Is such a repetition completely necessary? I think the authors may find a better set of sentences to expose their ideas on the matter. In the same paragraph one finds twice DSMC (which is not used anymore in the paper), twice MD (which is also never used after that), and DBM. The latter is used FOUR times in the immediate subsequent paragraph, and so on. Frankly, reading texts like the one we are dealing with results quite heavy, so the possible readers will lost interest immediately. To see the effect of such an excess, note that the meaning of MRT is lost in the paragraph before Eq (1). At this point, after facing a large amount of acronyms (that nobody is obligated to learn in order to read the paper), the reader has already forgotten what MRT means... although such an expression was introduced only a few lines above! On the other had, expressions like "NS model" are used without mentioning what the acronym means (in this case NS).
2.- Although section 2 is included to give a (very very short) introduction to the discrete Boltzmann method, such a section has no sense without explaining the reasons for using discrete models. I suggest to move section 2 to an appendix and to include instead a section2 where some of the reasons and motivations to include discrete mathematical models in the authors' approach. It is also expected an explanation about the limitations of the conventional continuous mathematical structures in modeling the behavior of shock waves.
3.- Section 3 is unclear, after reading it one gets no concrete idea about what the authors are looking for. It looks more than a collection of numerical results than the facing of a concrete problem with specific objectives.
4.- Section 4 contains the major contribution of the paper, which indeed is numerical. The exposition of results, commenting figures, is good. However, it seems to be disconnected from sections 2 and 3 since the latter are unclear, see items 2 and 3.
5.- Section 5, together with section 4, provides some insights about the novelty and soundness of the paper's contribution. However, the first 3 sections are fuzzy and unconnected with the last 2 ones.
I suggest to rewrite sections 1,2 and 3, in order to get a unified manuscript, where the ideas are exposed with clarity and the results are consistent with the model (and/or approach) used by the authors.
The authors must prepare a new version of the manuscript considering the previous items. After that, I will be happy to review the amended paper.
Reviewer 2 Report
This article presents a numerical study on characterization of non-equilibrium effects from both hydrodynamic and thermodynamic aspects, using DBM.
First, I have several questions on DBM:
There are many methods that aimed to fill the gap between N-S and Boltzmann but the authors are ignoring them.
1) how does it differ from LBM? which is also based on BGK.
2) how would it compare against GKS from Kun Xu’s group?
I don’t need comparison in terms of numerical simulation. I just need your professional comments on these different methods.
Second, I have some major concerns:
1) Most of the audience of Entropy are not familiar with DBM method. So a bit introduction is needed. Especially this paper contains almost zero information on DBM methodology, which makes this paper completely incomplete. Just take a look at the M_ij matrix, what is it? No explicit expression is presented. I don’t need a complete derivation because you would simply refer me to another 10 different papers. I just need this manuscript to be self-contained just for someone else who wants to reproduce your result.
2) When you discrete it in the velocity space, you end up with a delta approximation of the continuous velocity distribution function. This is common in many other techniques. But you didn’t explain how to compute the “du/dx” partial derivative term in equation (2). Do you have to compute those moments first, e.g., u,T, then compute derivative of them? As far as I can see, the equation (2) and (3) represent the “contribution from the difference between thermo and viscous diffusion”.
3) Look at the figure 2, there is no comparison against the exact solution (which is based on Euler equation). At least you should mention what kind of physics are you solving, what’s the governing equations, and how it differs from what most of the readers know, i.e., the exact solution from characteristic analysis. This becomes unclear why your results reflect the ground true physics or not. I know you can show some references that show the match between perhaps experiments and DBM but that doesn’t account for this paper. Otherwise, this paper would just look like you propose DBM with some hand tuned parameters and showed some colorful pictures. This is exactly what we should avoid in CFD community.
Therefore, I cannot recommend this paper for publication.
Round 2
Reviewer 1 Report
The faults of the previous work have been revised and corrected to produce a new manuscript, which is now coherent and clear in the exposition of ideas. The amended version of the manuscript seems to be free of technical errors. Indeed, it is clear now that the paper contributes to the applications of the discrete Boltzmann method in the study of shock waves. I have no additional comments or recommendations. The paper can be published in its present form.
Reviewer 2 Report
The authors have successfully addressed most of my concerns.
One more question is, are you willing to make the simple 1D code open-source? It will be beneficial for others to learn more about DBM. Note that you didn't answer directly on the "how the methods are different or similar to another type of Gas-Kinetic methods, which are also based on Boltzmann equation instead of N-S?". So it will be nice for other followers to figure out if you make your DBM code open-source.